# Evaluating the performance of automated detection systems for long-term monitoring of delphinids in diverse marine soundscapes

**Ellen L. White** [1]*, **Paul R. White** [2], **Jonathan M. Bull**[1], **Denise Risch**[3], **Susanna Quer**[4], **Suzanne Beck**[5]

1 School of Ocean and Earth Science, University of Southampton,Southampton, United Kingdom, 2 Institute of Sound and Vibration, University of Southampton, Southampton, United Kingdom, 3 Marine Science Department, Scottish Association of Marine Science, Oban, United Kingdom, 4 Renewables and Ecology Group, Marine Directorate, Scottish Government, United Kingdom, 5 Agri-Food and Biosciences Institute, Fisheries and Aquatic Ecosystems Branch, Northern Ireland, United Kingdom

* elw1d23@soton.ac.uk

## Abstract

There is an increasing reliance on passive acoustic monitoring (PAM) as a cost-effective method for monitoring cetaceans, necessitating robust and efficient automated tools for extracting species presence. This work compares two methods, one based on the 'off-line' analysis of raw PAM data, using Convolutional Neural Networks (CNNs), and the second based on in-situ detections, implemented within the C-POD. The C-POD is a rapid, low-cost choice for monitoring of odontocetes, while CNNs, requiring large efforts to train, are gaining traction within bioacoustics as they offer performance benefits above standard detection and classification tools. This work represents the first empirical comparison of a C-POD with a system using a CNN on recorded raw acoustic data for monitoring delphinids. The comparison is based on 3000 hours of PAM data, collected off the west coast of Scotland, using a collocated C-POD and SoundTrap acoustic recorder. Results show that the system using a CNN achieves an overall accuracy of 0.82, and an effectiveness (F1-Score) of 0.78 as a click detector, whilst the C-POD achieves scores of 0.71 and 0.62, respectively. The method employing a CNN provides a lower missed detection rate, with the C-POD failing to detect > 90% delphinid positive hours at one focal site. However, the C-POD offered a lower false-positive rate across all analysis sites. This work highlights the importance of incorporating the right automated tools for long-term species monitoring, as the C-POD offers high precision rates for click detections, while the CNN based system provides a robust approach to identifying seasonal and diurnal trends in long-term dolphin occurrence.

**Data availability statement:** A repository of data used in this manuscript can be found at: 10.5281/zenodo.15297395 This repository contains raw output files from the CNN, analysis files output from the CPOD.exe software and MATLAB analysis scripts. The raw acoustic files are too large for public repositories, please contact the lead author for access queries to the passive acoustic recordings.

**Funding:** This work was supported by the Natural Environmental Research Council [grant number NE/S007210/1]. The COMPASS project has been supported by the EU's INTERREG VA Programme, managed by the Special EU Programmes Body. The views and opinions expressed in this document do not necessarily reflect those of the European Commission or the Special EU Programmes Body (SEUPB).The European Commission, nor the SEUPB, played any role in study design, data collection and analysis, decision to publish, or the preparation of this manuscript'

**Competing interests:** The authors have declared that no competing interests exist.

## 1. Introduction

Effective long-term monitoring strategies for marine species are essential for evaluating their status and assessing responses to environmental change. The status and health of regional populations provide valuable insights into the broader environment, providing important metrics for conservation and assessing ecosystem health [1,2]. In order to make informed decisions about existing practices and develop new mitigations to effectively protect marine species, policymakers require reliable sound scientific information regarding species habitat use and distribution [3,4]. Passive acoustic monitoring (PAM) offers a cost-effective observation platform for gathering data over temporal scales of years to decades, used extensively for monitoring the distribution of a variety of marine species [5–7].

Acoustic data has yielded new and important insights into cetacean occurrence, movements and behaviour [7–13], by using acoustic cues, characteristic of the species of interest to quantify presence [14]. Dolphins, a common indicator species within the British Isles protected via Annex IV of the EU Habitats Directive [15] emit a diverse range of signals, employed in hunting, navigation and communication [15]. Their repertoires consist of highly directional transient signals between 20–140 kHz known as echolocation clicks, short pulsed transient signals with high repetition rates known as burst pulses (including squawks, screams and brays) [16], and tonal omni-directional, frequency modulated whistles, with a bandwidth between 2 and 35 kHz [15]. The structure and function of these signals differ within and between delphinid species, influenced by factors such as group size, group dynamic, behavioural activity, body size and phylogenetic relatedness [8,17–21].

To effectively use PAM for monitoring over time and space, practitioners can choose from a range of data collection approaches [22]. For studies monitoring species presence a choice must be made between on and offline systems. Off-line pipelines consist of storing archive raw acoustic data for postprocessing, which is resource-intensive but allows for a more comprehensive ecological analysis of the recovered data. In contrast systems can perform on-line detections in near real-time, archiving minimal data which characterises a signal when a detection is registered [23]. These systems offer massive data compression benefits and extend deployment times beyond those of a broadband recording system, however they lack flexibility with little to no ability to manually validate the reported detections. This can result in them being less suitable for studies where the species diversity and soundscape characteristics need to be investigated further.

The C-POD logger (Cetacean – Porpoise Detector, Chelonia Ltd UK) is a PAM system which performs on-line detections and has been popular in conservation research for small cetacean species for many years [22,24–27]. Although now superseded by a new model, the F-POD [28], the C-POD continues to be used as an integral tool within global monitoring programs for assessing spatio-temporal patterns of species occurrence [28–32] due to its capability for low-cost long-term continuous monitoring. Onboard the C-POD autonomously detects the presence of clicks between 20–160 kHz in real-time, storing summary data which is used to link detected events into species specific trains during post-processing. Proprietary

software containing a custom classifier (KERNO) classifies clicks into trains (regularly spaced series of similar clicks) based on their intensity, duration, frequency content and inter-click intervals (ICI), for which three classification filters are available: high-, medium-, and low-quality detections [33,34]. This offers a standardised simple post-processing chain which can yield information quickly after recovery. For commercial reasons detailed information is unavailable regarding the algorithms used in the C-POD.

Over the last decade, there has been a great increase in the availability and affordability of archival underwater acoustic recording systems which has led to a dramatic growth in their utilisation. In contrast to online systems many archival recording systems have limited continuous recording capabilities, limited by large data sizes and practicality of large battery packs. Practitioners may opt to record with a duty cycle to extend the maximum deployment time, but the resulting datasets only cover a fraction of the monitoring period. Collecting large volumes of acoustic data allows for various ecological questions to be explored regarding the species of interest and their acoustic habitat [35,36], but this scale presents significant analytical challenges. Comprehensive manual analysis of the recorded data is difficult, labour-intensive, and subjective, leading to significant inconsistencies among analysts when identifying signals within complex acoustic conditions [37–39]. This often results in non-standardized PAM data analysis, complicating the comparison of results between research groups. Automated tools which aid standardisation are therefore essential for efficiently gathering long-term species presence data in a timely manner for communication to policymakers and other researchers.

The use of signal processing methods to speed up the analysis of raw PAM data has a long history, but recently the advances in Machine Learning (ML) has greatly increased the potential for automation [40–43]. One widely adopted ML approach makes use of Convolutional Neural Networks (CNNs) which were originally developed for image processing, but have been used to classify acoustic data and have achieved high accuracies when applied to the task of detecting marine mammal vocalisations [40,41,44–46], including dolphin vocalisations within wideband PAM data recorded in variable marine environments [47–51], compared to other state of the art machine learning techniques [52].

An online system such as the C-POD offers an efficient standardised approach to monitoring odontocetes but lacks transparency and flexibility. In contrast, using raw PAM data allows for a more flexible approach to species monitoring but it can lack standardization, and offer incomplete temporal coverage. A significant challenge for CNNs is their ability to generalise and perform with high accuracy in unseen environments. Their ability to compete with the current standard processing pipeline across diverse marine soundscapes has yet to be demonstrated in the field of marine mammal acoustic classification. In this work we conduct the first empirical evaluation of the performances of a CNN, applied to broadband acoustic data recorded on a SoundTrap 300HF (Ocean Instruments), and the C-POD on the task of delphinid detection off the west coast of Scotland (Fig 1). An open-source multi-sound source CNN [49] is used without any network adaptation to the test data. We compare the output of both algorithms, across diverse acoustic conditions, against manually labelled recordings to evaluate their suitability in the standard processing toolkit for long-term delphinid monitoring.

## 2. Methods

This section outlines the methodology used, beginning with a detailed summary of the data acquisition process (Section 2.1), describing the analysis of acoustic data through each approach: a) manual, b) the CNN, and c) the C-POD (Section 2.2) and the evaluation metrics used in subsequent sections to assess the performance of each approach (Section 2.3).

### 2.1 Data Acquisition

This work used PAM data recorded within the COMPASS array (EU INTERREG COMPASS project), which consists of 8 moorings on the West of Scotland (Fig 1). Based on regional surveys [53–55] in this area the coastal waters (within 200m of shore) are important year-round marine habitats for the *delphinidae* family. The distribution of delphinid species one might expect to encounter in the region include long-beaked common dolphin (*Delphinus capensis*), bottlenose dolphin

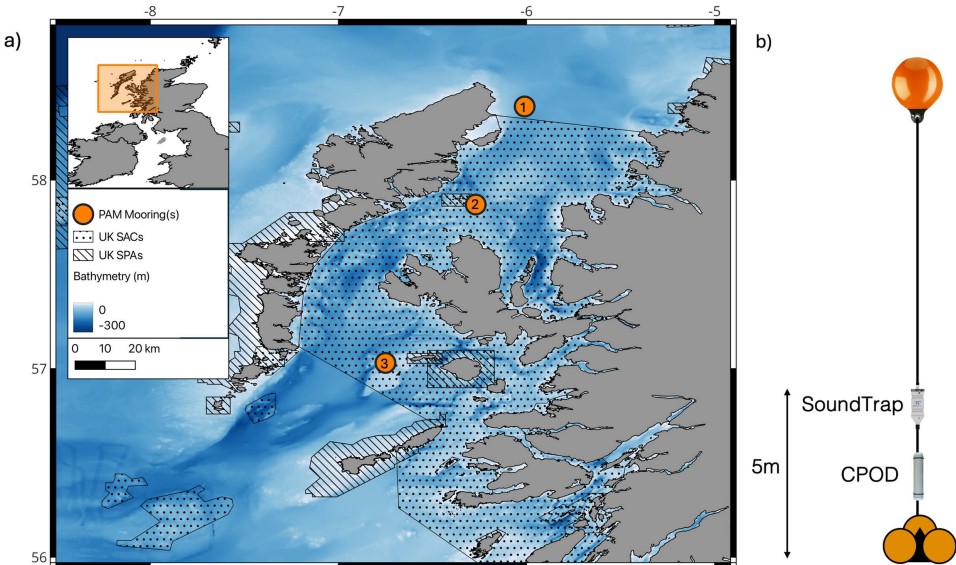

**Fig 1. a) Map of the COMPASS array acoustic monitoring sites used within this work located off western Scotland.** Acoustic moorings are numbered North to South; 1. Tolsta, 2. Shiant Isles, 3. Hyskier (Table 1). Marine protection designations are denoted on the map for Special Areas of Conservation (SACs) and Special Protected Areas (SPAs). The regional bathymetry is depicted by the colour scale (m), sourced from the General Bathymetric Chart of the Oceans (GEBCO), Country boundaries are sourced from geoBoundaries (https://doi.org/10.1371/journal.pone.0231866). b) Schematic depiction of each acoustic mooring consisting of a broadband recorder (SoundTrap 300HF, Ocean Instruments) and a C-POD logger positioned 5 m above the seabed.

(*Tursipos truncatus*), white-beaked dolphins (*Lagenorhynchus albirostris*), Risso's dolphin (*Grampus griseus*), Atlantic white-sided dolphins (*L. acutus*), killer whales (*Orcinus orca*) and long-finned pilot whale (*Globicephala melas*) [11,56–59].

Each mooring consists of a single omnidirectional broadband acoustic recorder, the SoundTrap 300HF (Ocean Instruments Ltd), located 5 m above the seabed (Fig 1). Audio data is sampled at 96 kHz and recorded on a 20/40 minutes on/off duty cycle which commences on the hour. A C-POD is positioned beneath the SoundTrap on the array (Fig 1b), monitoring continuously. The moorings were deployed simultaneously, so represent data from three different instruments of each type.

To compare the performance of the detection methods, we focused on data covering 3000 hours of deployment, from three recording sites: Tolsta, Hyskier and Shiant Isles (Fig 1, Table 1). Eight seasonal periods were chosen to evaluate

**Table 1.** Description of the acoustic data used within this work, recorded across the COMPASS array. Tolsta, Shiant Isles and Hyskier are the focal sites used to evaluate the performance of the methods. The seasonal periods refer to: J – January 24th – 31st, A – April 1st – 7th, Ju – July 1st – 7th, S – September 1st – 7th and N – November 17th – 23rd 2019, as well as three storm periods: St. G – Storm Gareth, St. H – Storm Hannah, and St.A – Storm Atiyah. Data gaps are present across sites, for this reason seasonal periods used vary between the sites.

| Site | Latitude | Longitude | Recorder Depth | Seasonal Period | Analysis Hours |
|------|----------|-----------|----------------|-----------------|----------------|
| 1. Tolsta | 58.39º N | −6.01º W | 100 m | J/ A/ Ju/ N St. G/ St. H/ St. A | 984 (226 Gb) |
| 2. Shiant Isles | 57.87º N | −6.27º W | 73 m | J/ A/ Ju/ S/ N | 864 (199 Gb) |
| 3. Hyskier | 57.03º N | −6.75º W | 55 m | J/ A/ Ju/ S/ N St. G/ St. H/ St. A | 1152 (265 GB) |
| | **Total Analysis Hours** | | | | 3000 (600 Gb) |

detector performance. Five one-week periods were selected to cover the annual cycle, with three additional four-day periods chosen to coincide with storm events. The five one-week periods were selected to be representative of seasonality across the year; January 24th – 31st, April 1st – 7th, July 1st – 7th, September 1st – 7th and November 17th – 23rd 2019. No acoustic data used within this work was included within the CNNs training data (White *et* al., 2022), however we do note that the training set consisted of acoustic data recorded at Tolsta during November 2017. In addition, three periods of four days were selected to coincide with low-pressure systems, corresponding to times when a named storm event was present in the area. These periods were identified using the UK's MetOffice Integrated Data Archive System (MIDAS) weather records for 2019 [60]. The selected storms were Storm Gareth (March 11th – 14th 2019), Storm Hannah (April 25th – 28th 2019), and Storm Atiyah (December 7th – 10th 2019).

After decompression the total acoustic data for input to the CNN is 690 Gb (Table 1).

## 2.2 Analysis of acoustic data

The analysis of the performance of both systems is based on detection positive hours (dph), i.e., an hour is dolphin positive if any acoustic detection of a dolphin is made within that hour. The following describes how the number of dph were determined manually, from the CNN, and from the C-POD.

### a) Manual detection

Each of the 20-minute broadband acoustic files was reviewed visually (spectrograms) and aurally in Audacity (version 3.0.02, 2021) to identify the hourly presence/absence of delphinid click trains and whistles. Spectrograms were viewed consecutively in 10-second windows, with data displayed linearly between 0–48 kHz using a Hanning window of size 2048, a 50% overlap, and a dynamic range of 80 dB. Delphinid clicks were identified based on the temporal and frequency characteristics of broadband signals within the data, including the frequency range and inter-click-interval (ICI).

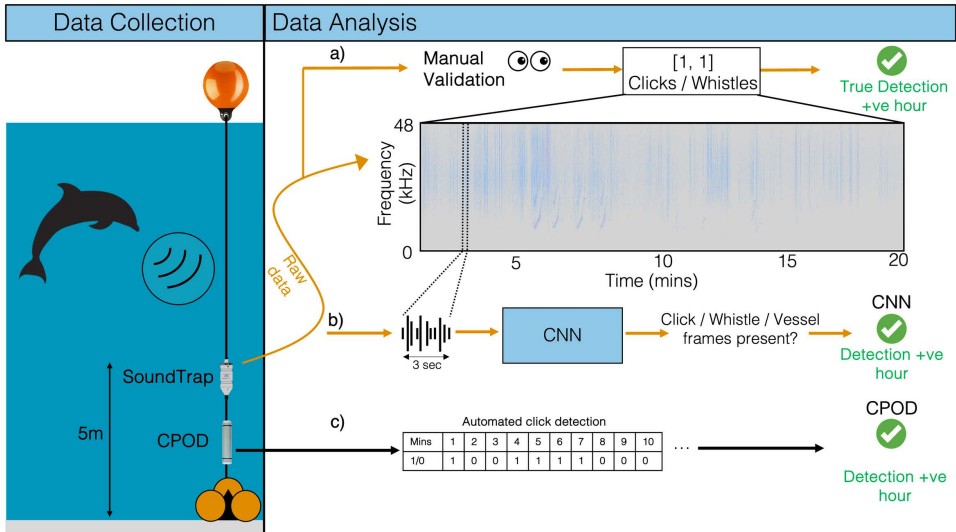

**Fig 2. Schematic depiction of the data collection and subsequent analysis workflow used for the determination of delphinid detection positive hours (dph) by a) Manual, b) CNN, and c) CPOD detection.** Manual analysis applies a positive or negative label for the presence of delphinid clicks and delphinid whistles. Audio data is fed through the CNN which outputs one label per input frame (3 seconds); Ambient noise, delphinid whistle, delphinid click or vessel noise, and the total frames per class are summed per file to provide hourly presence/absence of clicks. The C-POD outputs detection positive minutes for delphinid clicks, for each minute of the recording hour, with the total sum of positive minutes converted to positive or negative detection hour.

A click train was identified as present if ≥3 clicks were present in series during the 20-minute recording period (Fig 2). A secondary label was assigned if any whistles were detected during the manual review. The output array does not contain counts, but an hourly record of the presence of delphinid vocalisations.

## b) CNN detection

The details of the network architecture for the CNN used in this task can be found in White *et al* [49], with additional information available in the Supplementary Material in S1 File. The original network, built upon the EfficientNet B0 architecture, was trained on acoustic data recorded in the COMPASS array during 2017 and 2018. The acoustic data used for training the system were collected in completely different years from one of the sites in this study, and there was only one site in common, specifically Tolsta, see White et al., (2022). This ensured that the CNN operated on broadband data outside of the training distribution, an unseen dataset.

Each 20-minute file of acoustic data, from the total 600Gb dataset, was split into 400 frames of 3-second duration. Each frame is converted to an input image by computing three individual spectrograms using three FFT windowing sizes: 1024, 2048 and 4096. The spectrograms use a Hanning window with 50% overlap and the amplitude is normalised such that the maximum value is 0 dB. The spectrograms are resized to 224x224 pixels and stacked to form a three-channel tensor, representing the RGB input image [49]. The network classifies each frame into one of four labels: ambient noise, delphinid whistles, delphinid clicks (click trains and burst pulses) or vessel noise, outputting a labelling file listing the labels in temporal order, along with the model's probability score used to assign the label. The probability score per detection ranges between 0 and 1, where 1 implies 100% confidence in the detection.

Overall, the CNN took 30 hours to process the broadband acoustic data for all sites and season, averaging 1.5 hours per week of data. Across the three sites this amounted to 10.5 hours for Tolsta, 7.5 hours for Shiant Isles, and 12 hours for Hyskier.

The CNN was trained to detect more than a single component of the delphinid vocal repertoire, accounting for both echolocation clicks and whistles. Assigning only one label per frame, the model was trained to label in a hierarchy, where whistles were prioritised over clicks [49]. As a result, missed click detections may be a result of the model identifying whistles when both signals occur in tandem. For each file (and hence the hour) a dph is determined for each class if the model output any frames with the respective label (Fig 2). In post-processing a threshold can applied to the CNN output score array allowing the user to control the trade-off between false alarms and missed detections when computing the dph. This ensures that only detections above the threshold are considered when determining dphs. A threshold of 0.9 is applied in Section 3.5 of this manuscript.

## c) C-POD detection

The C-POD detection files were analysed using proprietary software provided by Chelonia Ltd. The KERNO classifier, the base classifier provided for C-POD delphinid detection, was used to identify delphinid click trains. The software provides filter settings for the click detection of 'High', 'Moderate' or 'Low' quality trains [25]. Click trains of all three quality levels were accepted and any hour in which a click train was detected was labelled as being dph (Fig 2). Total processing time to extract detection positive hours from the C-POD was one hour for all sites and seasons. The total size of extracted data from the C-POD was 2Mb, totalling 650 Kb per site for all analysis weeks. This is a significant reduction in data volume compared the raw acoustic data necessary for the CNN (Table 1).

Note that the C-POD continuously recorded for the full hour, in contrast to the PAM data which only captures the first 20-minutes within each hour. This is a practical advantage that the C-POD offers over collecting raw PAM data, so it is reasonable to maintain this benefit of C-PODs, although the ground-truth labels account for only one third of the C-PODs recording period. As a result of environmental noise or movement in the water column the C-POD may switch off, or cease monitoring for a short period of time (< 1minute), known as 'time-lost'. The percentage of time lost averages <10% per hour for the analysis data, the breakdown of this per site and season is reported in the supplementary material in S1 File.

## 2.3 Evaluation

To quantify the performance of each detection method Accuracy (*A*), Precision (*P*), Recall (*R*) and the F1-score were computed (equations 1–4). We note that Recall is also referred to as the True-Positive Rate (*TPR*) within this manuscript. These calculations use the manual labels as ground truth.

$$A = \frac{N_{TP} + N_{TN}}{N_{all}}, \tag{1}$$

$$P = \frac{N_{TP}}{N_{TP} + N_{FP}}, \tag{2}$$

$$R = \frac{N_{TP}}{N_{TP} + N_{FN}}, \tag{3}$$

$$F1 = \frac{N_{TP}}{N_{TP} + \frac{1}{2}(N_{FP} + N_{FN})} \tag{4}$$

Where $N_{TP}$ is the number of true positives, $N_{TN}$ is the number of true negatives, $N_{FP}$ is the number of false positives, $N_{FN}$ is the number of false negatives and $N_{all}$ is the total number of frames in the evaluation set. The false-positive rate (*FPR*) is the proportion of the positive detections that are incorrect, and the false-negative rate (*FNR*) is defined equivalently for the negative detections. The false negative rate is the rate of missed detections and is equal to 1 – Recall:

$$FPR = \frac{N_{FP}}{N_{FP} + N_{TN}}, \tag{5}$$

$$FNR = \frac{N_{FN}}{N_{FN} + N_{TP}} = 1 - R. \tag{6}$$

## 3. Results

This section evaluates the performance of the C-POD and the CNN, applied to broadband acoustic data, in detecting delphinid presence across diverse underwater soundscapes. This analysis is based on 3,000 hours of acoustic data collected throughout 2019 in western Scottish waters. First, we conduct an overall comparison of click detections (Section 3.1). The output of each detector is then examined to identify temporal and seasonal patterns (Section 3.2). Sections 3.3 and 3.4 compare detector performance across varying ambient noise conditions using ROC curves and precision-recall metrics. Finally, Section 3.5 incorporates whistle detections output by the CNN to evaluate the benefits of monitoring delphinids using multiple signal types.

### 3.1 Click detection evaluation

The results of the manual analysis are summarized in Table 2, with empty cells representing data gaps, see Table 1. Echolocation click detections were evaluated with respect to the manually validated labels assigned to each of the 20-minute files, reporting a total of 452 dph at Tolsta, 313 dph at Hyskier and 298 dph at Shiant Isles (Table 2). The overall total dph is 1063, with 1937 hours without any delphinid detections made, for an imbalanced dataset of roughly 2:1 in favour of no detections. Table 2 also highlights seasonal trends in animal presence, with higher dph recorded in

November, and the fewest in April. With respect to site, Tolsta experiences the greatest proportion of dph (46%) and Hyskier the lowest (27%), with the effect of storms most notable at Hyskier with only 19% of hours positive detections for delphinid presence across the three periods (Gareth = 7%, Hannah = 5%, and Atiyah = 35%, Table 2).

Table 3 provides details of the performance of the CNN and the C-POD, using the manual labels in Table 2 as ground-truth, averaged across the sites. Analysis of the acoustic data by the C-POD was rapid, less than 1 hour for the whole dataset, with the CNN averaging a speed of 1.5 hours per week of data, when using MATLAB 2023b on a MacBook Pro 2023, with an M3 Pro chip running Sonoma 14.1.

From Table 3 it is evident that the C-POD's performance is characterised by a low level of false positives (overall 2% for the C-POD compared to 7% for the CNN), which results in a correspondingly high precision for the CPOD (0.96, Table 3).

**Table 2. The total true positive ($N_{TP}$) and true negative ($N_{TN}$) dph for delphinid clicks during the manual analysis of each temporal period and site.** Temporal periods are defined as: Jan – January (24th – 31st), Apr – April (1st – 7th), Jul – July (1st – 7th), Sep – September (1st – 7th), Nov – November (17th – 23rd), St. G – Storm Gareth, St. H – Storm Hannah, and St. A – Storm Atiyah.

| Site | | Total Hours | Overall | Jan | Apr | Jul | Sept | Nov | St. G | St. H | St. A |
|---|---|---|---|---|---|---|---|---|---|---|---|
| Tolsta | $N_{TP}$ | 984 | 452 (46%) | 122 | 33 | 56 | | 116 | 33 | 22 | 70 |
| | $N_{TN}$ | | 532 | 70 | 135 | 112 | | 52 | 63 | 74 | 26 |
| Hyskier | $N_{TP}$ | 1152 | 313 (27%) | 33 | 10 | 52 | 57 | 115 | 7 | 5 | 34 |
| | $N_{TN}$ | | 839 | 159 | 158 | 116 | 111 | 53 | 89 | 91 | 62 |
| Shiant Isles | $N_{TP}$ | 864 | 298 (34%) | 39 | 43 | 46 | 93 | 77 | | | |
| | $N_{TN}$ | | 566 | 153 | 125 | 122 | 75 | 91 | | | |

**Table 3. Performance metrics for the CNN and C-POD for each analysis period, with respect to the manually validated data at Tolsta, Hyskier and Shiant Isles combined.** Metrics reported are the number of true-positives (*NTP*), number of true-negatives (*NTN*), number of false-negatives (*NFN*), number of false-positives (*NFP*), accuracy (*A*), precision (*P*), recall (*R*), *F*1-score and the false-positive rate (*FPR*).

| | Period | $N_{TP}$ | $N_{TN}$ | $N_{FN}$ | $N_{FP}$ | A | P | R | F1 | FPR |
|---|---|---|---|---|---|---|---|---|---|---|
| **CNN** | **Overall** | **978** | **1459** | **430** | **133** | **0.82** | **0.89** | **0.70** | **0.78** | **0.07** |
| | Jan | 242 | 234 | 64 | 36 | 0.83 | 0.87 | 0.80 | 0.83 | 0.13 |
| | Apr | 133 | 291 | 51 | 29 | 0.84 | 0.82 | 0.73 | 0.77 | 0.09 |
| | July | 164 | 250 | 78 | 12 | 0.82 | 0.93 | 0.68 | 0.78 | 0.04 |
| | Sep | 90 | 157 | 65 | 24 | 0.74 | 0.79 | 0.58 | 0.67 | 0.13 |
| | Nov | 253 | 138 | 97 | 16 | 0.78 | 0.94 | 0.73 | 0.82 | 0.10 |
| | St.G | 25 | 145 | 15 | 7 | 0.89 | 0.78 | 0.62 | 0.70 | 0.04 |
| | St.H | 12 | 158 | 15 | 7 | 0.89 | 0.63 | 0.45 | 0.52 | 0.04 |
| | St.A | 59 | 86 | 45 | 2 | 0.76 | 0.97 | 0.57 | 0.72 | 0.02 |
| **CPOD** | **Overall** | **750** | **1380** | **838** | **36** | **0.71** | **0.96** | **0.47** | **0.62** | **0.02** |
| | Jan | 217 | 226 | 128 | 5 | 0.77 | 0.98 | 0.62 | 0.76 | 0.02 |
| | Apr | 131 | 290 | 78 | 5 | 0.84 | 0.96 | 0.63 | 0.76 | 0.01 |
| | July | 144 | 233 | 122 | 5 | 0.75 | 0.96 | 0.54 | 0.69 | 0.02 |
| | Sep | 77 | 124 | 131 | 4 | 0.59 | 0.95 | 0.37 | 0.53 | 0.03 |
| | Nov | 138 | 109 | 251 | 6 | 0.49 | 0.95 | 0.35 | 0.51 | 0.05 |
| | St.G | 7 | 151 | 33 | 1 | 0.82 | 0.87 | 0.18 | 0.29 | 0.01 |
| | St.H | 2 | 161 | 25 | 4 | 0.84 | 0.33 | 0.07 | 0.12 | 0.02 |
| | St.A | 34 | 86 | 70 | 2 | 0.62 | 0.94 | 0.32 | 0.48 | 0.02 |

However, this is paired with a low recall score of 0.47 and a large number of missed detections, specifically a *FNR* of 53% compared to 30% for the CNN. The F1 score, a metric which balances precision and recall, is 0.78 for the CNN overall, ranging from 0.52–0.82, compared to 0.62 for the C-POD overall for which scores ranged between 0.12–0.76.

In terms of accuracy, the CNN achieves 82% over the entire dataset, versus 71% accuracy by the C-POD. The spread of accuracies for the two systems was 74–89% for the CNN and 49–84% for the CPOD, with both systems outputting their lowest accuracy in September, and the highest accuracy during Storm Hannah, when the F1 scores were lowest (Table 3). This apparent contradiction arises as a result of the extreme data imbalance during Storm Hannah where only 14% of hours were detection positive. Such a large imbalance makes it possible to have high accuracies and a low F1-Score where systems can class every hour as negative and still achieve 86% accuracy, but an F1-Score of 0.

### 3.2  The influence on observed patterns in the data

To obtain finer detail on the robustness of each algorithm for describing temporal occurrence we summed the number of times, for each hour of the day, that hour was detection positive, computed across each seasonal period, per site (Fig 3). The manual labelling indicates that delphinids are most commonly detected in January and November, with both months presenting a clear diurnal pattern with the majority of detections occurring between 00:00–04:00 and 20:00–24:00, corresponding to hours of darkness. The diurnal pattern occurs across all analysis periods but is most evident when detections are high, in January and November, with the exception of January at Shiant Isles (Fig 3).

Both the CNN and C-POD identify the temporal patterns in animal presence described by the manual labels with varying degrees of success (Fig 3). A Pearsons correlation coefficient is used to quantify the temporal distributions of the CNN

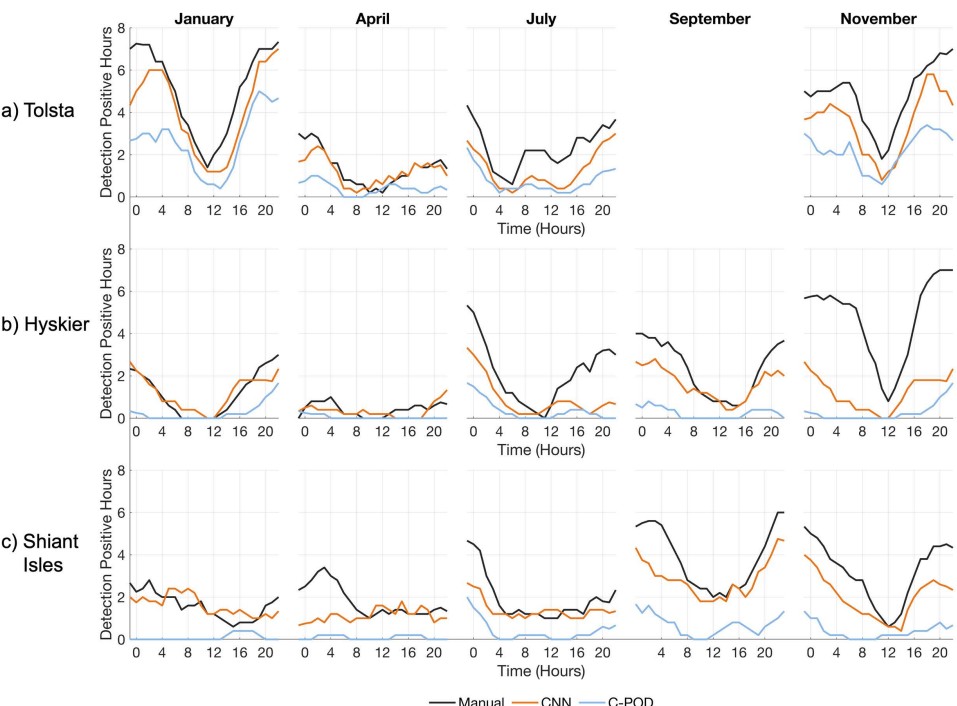

**Fig 3. The total number of delphinid positive hours per hour across each analysis period, for the CNN (orange line), the C-POD (blue line), and manual labels (black line) at a) Tolsta, b) Hyskier, and c) Shiant Isles.** The data has been smoothed using a running mean of 5 data points. The CNN provides an accurate but conservative depiction of seasonal and diel patterns in dolphin occurrence, highlighting the advantages of using it as a monitoring tool. The C-POD identifies temporal variability at Tolsta but poorly reflects the daily patterns at the other two locations.

and C-POD to the manual labels, averaged over the five analysis periods. The algorithms both output the highest mean correlation scores at Tolsta, with the CNN scoring 0.78, with scores ranging from 0.83 in January to 0.72 in November and the C-POD scoring 0.68, ranging between 0.77 in January and 0.63 in April. For both algorithms correlation was weakest at Shiant Isles, with a mean correlation coefficient of 0.45 for the CNN and 0.24 for the C-POD. Details of the Pearsons correlation analysis can be found in the Supplementary material in S1 File.

Diurnal trends are obscured for the C-POD as a result of high false negative rates, underestimating the number of dph which leads to extended periods of the C-POD reporting no detections in contrast to the manual labels. This is particularly clear in Fig 3 for Shiant Isles during January, and April at Hyskier. During the highest densities of dolphin activity, during January and November at Tolsta, the C-POD outputs a high *FNR* but captures the overall diurnal trend in animal presence (Fig 3). Both algorithms are able to identify that dolphin presence increases throughout the year at Hyskier and Shiant Isles, with a higher proportion of dph in September and November (Fig 3).

### 3.3 Overall performance comparison

To understand how the ambient soundscape can affect detector performance we evaluated both algorithms during the three storm events (Table 4, Fig 4). Performance metrics are compared between periods of storm conditions and the remaining analysis periods (hereafter referred to as non-storm conditions). The overall *TPR* for the CNN is 0.31 higher than the C-POD during storm events, and 0.40 higher during non-storm conditions. Although identifying fewer detections, the C-POD has a notably lower *FPR*, 0.02, for both conditions, compared to the CNN, with a larger *FNR of 0.80* in non-storm conditions and 0.75 during storm conditions.

In general, detection methods offer a facility to allow the user to control the trade-off between the *TPR* and the *FPR* [61]. For a fair comparison of the performance of any two detector systems, one needs to consider configurations when the two systems have some suitable form of equivalence. For the CNN *TPR* and *FPR* can be measured for different operating thresholds to construct a receiver operating characteristic (ROC) curve and compare that to the point which represents the C-POD's performance. Such a plot is shown in Fig 4 a) with Fig 4 b) showing a similar related curve, the precision-recall curve, in the storm and non-storm conditions.

The CNN outperforms the C-POD by achieving a higher *TPR* for the same false-positive rate of 0.02 (Fig 4a) and the CNN precision is greater for the same recall scores as the C-POD in both storm and non-storm periods (Fig 4b).

### 3.4 Variation between sites

ROC and precision-recalls are computed for each site during storm and non-storm conditions, illustrated in Fig 5.

**Table 4. Performance metrics for the CNN and the C-POD during periods of storm conditions, and non-storm conditions.** Storm conditions include Storm Hannah, Gareth and Atiyah, No Storm conditions include January, April, July, September and November (see Table 2). Metrics reported are the true-positive rate (*TPR* = Recall), false-positive rate (*FPR*), false-negative rate (*FNR = 1 − TPR*), and the F1-score.

| Site | Algorithm | Storm | | | | No Storm | | | |
|---|---|---|---|---|---|---|---|---|---|
| | | *TPR* | *FPR* | *FNR* | *F1* | *TPR* | *FPR* | *FNR* | *F1* |
| Overall | CNN | 0.56 | 0.04 | 0.44 | 0.68 | 0.60 | 0.08 | 0.40 | 0.70 |
| | C-POD | 0.25 | 0.02 | 0.75 | 0.39 | 0.20 | 0.02 | 0.80 | 0.33 |
| Tolsta | CNN | 0.62 | 0.04 | 0.38 | 0.74 | 0.69 | 0.03 | 0.31 | 0.80 |
| | C-POD | 0.34 | 0.04 | 0.66 | 0.49 | 0.39 | 0.04 | 0.61 | 0.55 |
| Hyskier | CNN | 0.41 | 0.04 | 0.59 | 0.51 | 0.60 | 0.06 | 0.40 | 0.69 |
| | C-POD | 0.00 | 0.00 | 1.00 | 0.00 | 0.07 | 0.01 | 0.93 | 0.13 |
| Shiant Isles | CNN | | | | | 0.51 | 0.12 | 0.49 | 0.59 |
| | C-POD | | | | | 0.12 | 0.01 | 0.85 | 0.12 |

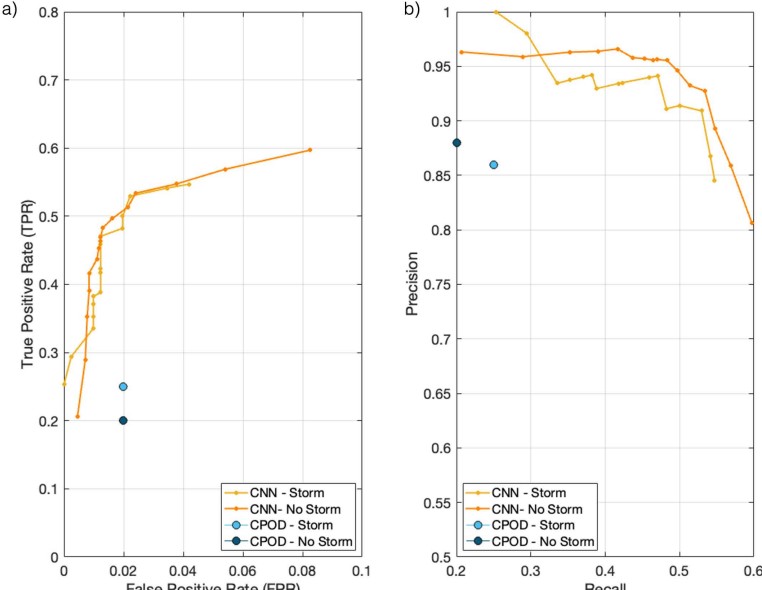

**Fig 4. Performance comparison of both the CNN and C-POD using a) a receiver operating curve, and b) a precision recall curve, to evaluate their efficiency under storm and non-storm conditions.**

At Tolsta the CNN produces a higher rate of true positive detections, with a comparative false-positive rate to the C-POD in both weather conditions (Table 4, Fig 5a,b,c). The influence of storm conditions appears to affect the performance of the C-POD and CNN to a similar degree, with the difference in F1-scores between the storm and non-storm periods 0.06 for both algorithms, Table 4.

At Hyskier the C-POD fails to detect any of the dph during storm conditions, and only correctly identifies 7% of the dph in non-storm conditions, which results in a low *FPR* impacting the precision and *TPR* scores (Table 4, Fig 5b). We note the C-POD reported high levels of time lost during Storm Gareth, limiting the proportion of time available for detecting dolphins (See Supplementary Material in S1 File). The analysis at Shiant Isles is restricted to non-storm periods only, however the CNN continues to outperform the C-POD as an overall detector. The CNN outputs a notably high FPR of 0.12 compared the C-PODs 0.01, however the CNN outputs a true-positive rate of 0.50 which is significantly higher than the C-PODs true-positive rate of 0.12 (Table 4, Fig 5c), indicating a difficult soundscape for classification.

Both detection algorithms are affected by location and ambient soundscape conditions, affecting their ability to robustly detect delphinid activity in long-term PAM datasets.

### 3.5 Performance of the CNN based on both clicks & whistles

Here we assess the model's overall performance with respect to both whistle and click detections (Fig 6), something which cannot be achieved with the C-POD. The CNN used in this work generates a single label for each frame, labelling it as either, delphinid whistle, delphinid click, vessel noise or ambient noise. Of these, both of the first two classes indicate delphinid presence, in this section we consider the impact of including data from the whistle class as well as the click class (Fig 6). In this case a dph was declared if either a whistle or click, or both signals were detected within a 20-minute file. A threshold of 0.90 was applied to both whistle and click detections output by from the CNN.

The inclusion of the whistle class does not dramatically change the temporal trends of delphinid presence but increases the number of overall hourly detections across each focal site (Figs 3 & 6). Using both whistles and clicks we find

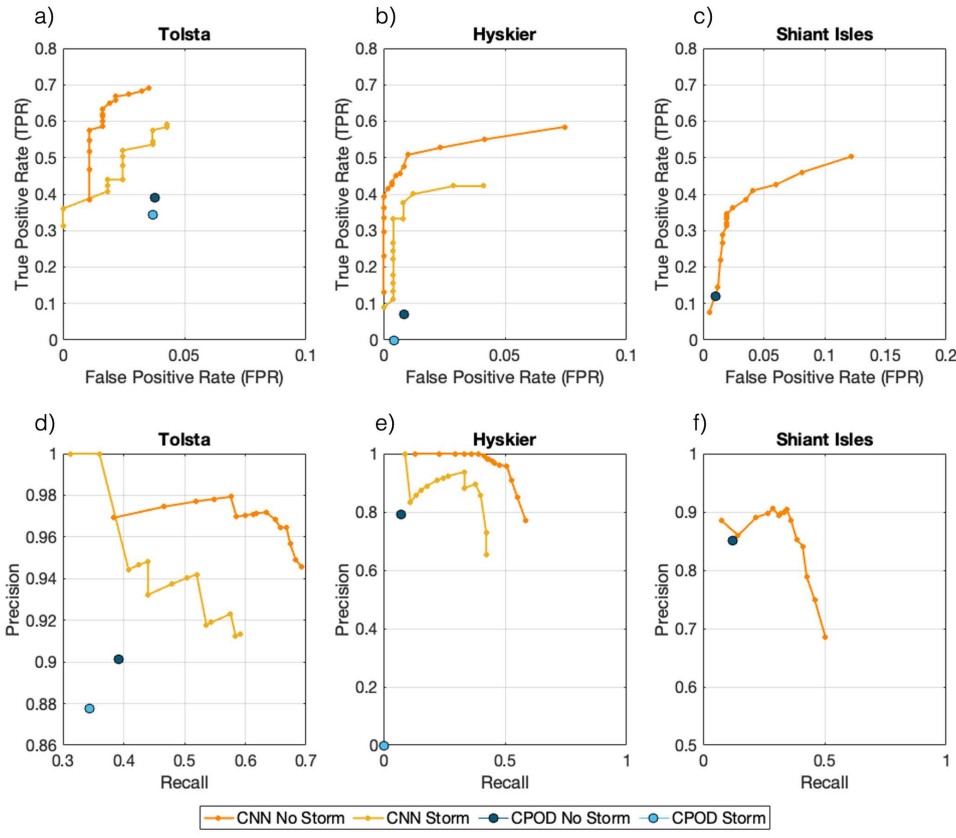

**Fig 5. Performance comparison of both the CNN and C-POD to evaluate their efficiency under storm and non-storm conditions at each of the focal sites: Tolsta, Hyskier and Shiant Isles.** Performance is evaluated using a receiver operating curve (a, b & c), and a precision recall curve (d, e, f). The CNN metrics are higher than that of the C-POD at each site for both storm and no-storm conditions. Note that the axis scales are not consistent across the plots.

delphinids occur in higher densities in the waters off Tolsta, particularly during January and November, with a decrease in presence between 12:00–16:00 (Fig 6), consistent with the click only analysis (Fig 3). Including whistles in the detectors output improves correlation to the manual labels, with the Pearsons coefficient improved when dolphin densities are high (see supplementary material in S1 File).

Comparing the true detection hours between the CNN and the manual labels, the best performance occurred at Tolsta, with correlation coefficients above 0.75 for each site, excluding April. During April the CNN outputs 319 more positive hours across the three moorings (Fig 6 & Table 5), with the biggest difference found at Hyskier (Table 5). At Shiant Isles the diurnal trend output contrasts with the manual analysis, where 115 more positive hours are detected, occurring mainly during January and April (Fig 6).

## 4. Discussion

Acoustically reliable methods of automated signal detection, robust to fluctuations in seasonal and regional ambient noise levels, are necessary for effective automated pipelines for monitoring marine populations, particularly in environments that have designated protection status. Here we compare an approach based on processing raw PAM data post-processed using a multi-sound source CNN, which is trained to detect delphinid signals across a wide frequency range (0–48 kHz), with a data logger, the C-POD, to evaluate their utility in the long-term management and conservation of dolphins.

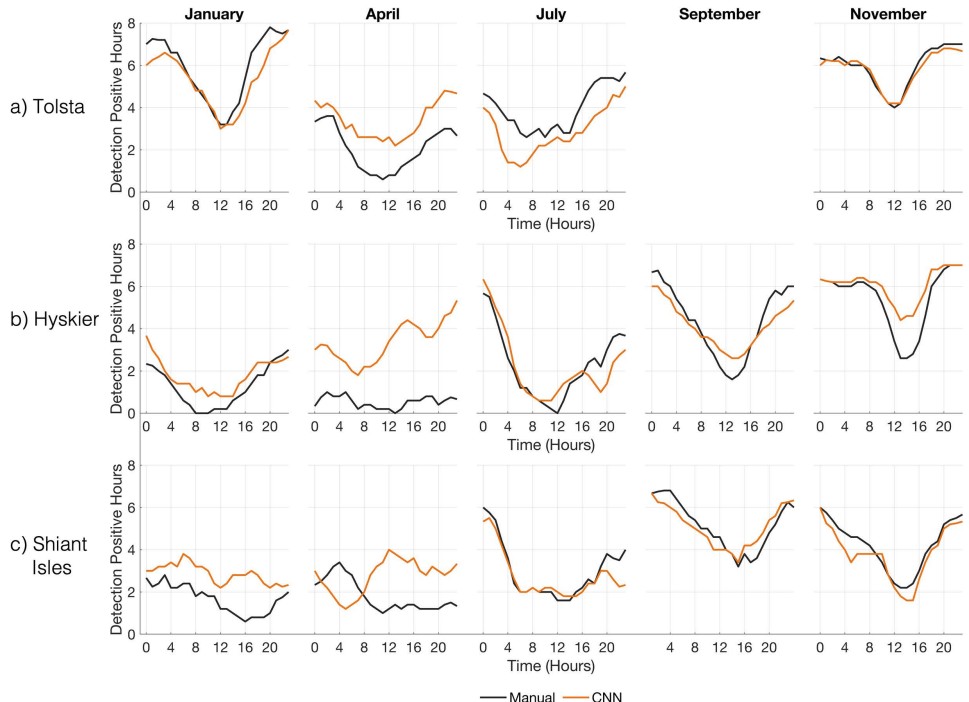

**Fig 6. The total number of delphinid positive hours per hour across each analysis period at a) Tolsta, b) Hyskier, and c) Shiant Isles, for both whistle and click detections for the Manual labels and the CNN.** To ensure a fair comparison, CPOD detections are not included here as they only account for echolocation clicks. All data has been smoothed using a running mean of 5 data points.

**Table 5. Total delphinid positive hours overall, and per site, for the manual labels, CNN and the CPOD.** The manual labels and CNN detections account for both whistle and click detections, with the C-POD only including click positive hours. Inclusion of the C-POD here is not to compare results to that of the CNN, but to demonstrate the CNNs performance with multi-signal analysis.

|  | Overall | | | Tolsta | | | Hyskier | | | Shiant Isles | | |
|---|---|---|---|---|---|---|---|---|---|---|---|---|
|  | **Man** | **CNN** | **CPOD** | **Man** | **CNN** | **CPOD** | **Man** | **CNN** | **CPOD** | **Man** | **CNN** | **CPOD** |
| Overall | 1361 | 2640 | 258 | 586 | 703 | 192 | 393 | 1097 | 25 | 382 | 840 | 41 |
| Jan | 214 | 512 | 71 | 142 | 146 | 62 | 31 | 183 | 7 | 41 | 183 | 2 |
| Apr | 105 | 424 | 13 | 49 | 108 | 10 | 13 | 158 | 1 | 43 | 158 | 2 |
| Jul | 223 | 428 | 37 | 95 | 94 | 18 | 56 | 167 | 9 | 72 | 167 | 10 |
| Sep | 228 | 328 | 23 |  |  |  | 104 | 164 | 6 | 124 | 164 | 17 |
| Nov | 374 | 485 | 64 | 142 | 149 | 53 | 130 | 168 | 1 | 102 | 168 | 10 |
| St. H | 47 | 142 | 8 | 40 | 58 | 8 | 7 | 84 | 0 |  |  |  |
| St. G | 45 | 143 | 6 | 35 | 65 | 5 | 10 | 78 | 1 |  |  |  |
| St. A | 125 | 178 | 36 | 83 | 83 | 36 | 42 | 95 | 0 |  |  |  |

There were some basic differences in the operation of each of the detection methods. C-PODs monitor the environment continuously throughout their deployment, while the raw PAM recordings were restricted to only capture data for the first 20 minutes in every hour to maximise storage capacity. This could potentially explain some apparent false positives on the C-POD compared to the manual annotations: if an animal is present only in the last 40 mins of an hour the PAM system will not record that animal's presence, whereas the C-POD might detect it and the event be classified as false positive. Our results show that the *FPR* for the C-POD is extremely low (4% at its highest, see Table 4) and so this appears

not to have adversely affected the C-PODs evaluation. The criteria for each system to register a detection also differs. The black-box nature of the CNN means that one cannot in detail understand the rationale behind detections, while the C-PODs operation is opaque for commercial reasons.

The CNN detected higher numbers of delphinid positive hours than the C-POD, for all analysis periods, with the degree of discrepancy between the two detectors varying seasonally for echolocation clicks (Section 3.1–3.5). The CNN achieved an overall accuracy of 82%, with performance varying across different ambient noise environments (63% – 97%) and required a relatively short analysis time (1.5 hours per week) compared to manual labelling. The CNN reported low false-positive rates (2% – 13%), and overall high precision and recall scores (Table 3), despite using unseen acoustic data. Seasonal and diel patterns in animal presence identified during the manual analysis were demonstrated by the CNN using a single sound source (Fig 3, Supplementary Table 1 in S1 File) highlighting the model as a reliable indicator of dolphin occurrence throughout the year, suitable for long-term monitoring within variable ambient noise conditions. Combining the detection of clicks and whistles improved the CNNs performance at identifying seasonal and diurnal detection of dolphin activity, demonstrating the benefit of applying a multi-sound source algorithm to PAM datasets. The performance of the C-POD balanced across evaluation metrics is lower for all focal sites than the CNN (Figs 4–5), however we note the sensitivity of this system. The *FPR* are consistently low for all ambient conditions, despite high rates of missed detections. For research and management questions where sensitivity is the most important metric, this detector is the suitable choice.

A key benefit of using a CNN on raw acoustic data is that the model can be fine-tuned to the dataset at hand. Previous work found that fine-tuning a CNN to the ambient soundscape in which it will be deployed can improve the performance of signal detection by 30% with very little additional training data [51]. For this work the model was deployed on unseen data, but by allowing the network to learn from a small subset of the raw PAM data we can enhance model performance and increase detection probability scores [51]. When applying a threshold of 0.90 we improved the model's ability to detect whistles and clicks in line with the manual analysis, reducing tonal false positives by 71%. To incorporate this model within the marine management toolkit, we recommend users make use of both thresholding and fine-tuning to ensure the highest detection sensitivity, acquiring robust long-term trends in animal presence per site. Where fine-tuning is not possible, we demonstrate confidence in the model used 'out-of-the-box', outperforming the current choice in long-term dolphin monitoring. We advise users to make use of the CNN output on a sub-sample of data to visualise and explore signals which lead to missed detections, or false positives, ensuring that any efforts to fine-tune the network to the study site are grounded in robust data exploration. This is a notable strength of any CNN pipeline over a data logger, the architecture can be overfit to any dataset for a boost in performance. However, to fine-tune a model there is a computational skillset required which may be too complex for some users to make use of, a barrier that is not present with a data logger.

The very low rate of false detections from the C-POD was combined with a failure to identify between 62–93% of the manually identified dphs for each individual site, while using the least conservative detection filters available. Our results are consistent with existing studies which have compared C-POD performance to other automated approaches (e.g., PAMGuard, [62]), concluding that although the C-POD produces very low false-positive rates, it under reports delphinid presence [32,33]). Dolphin echolocation clicks occur between 20–140 kHz but research on C-POD detection ranges found the sensitivity of the logger to be weakest below 80 kHz [31]. The reduction in sensitivity coupled with adverse ambient noise conditions presents a difficult operational environment for the C-POD, which must be considered when selecting a tool for long-term species monitoring. We must note that the C-POD has been superseded by the F-POD (Full waveform capture POD) [28] and so the issue of missed detections may be reduced if studies can acquire the new system.

The two methods are compared using ROC and precision-recall curves, allowing the assessment to be based on conditions when the *FPR* or recall for two systems has been equalised. From that analysis, we conclude that the CNN achieves better *TPR* than the C-POD when the *FPR* is equalised and higher precision than the C-POD for the same recall.

The CNN output a high rate of false-positives when accounting for both whistles and clicks, particularly Hyskier and Shiant Isles and during April (Fig 6). Manual annotation of this season revealed a continuous presence of an anthropogenic chirp signal, with an increasing frequency structure, present between 1–15 kHz (Supplementary Figure 3 in S1 File). The chirp's structure is remarkably similar to a dolphin tonal call, occurring within adjacent frequency bins with a similar temporal duration. The signal varies in signal-to-noise ratio but is present between 40–55% of analysis hours in April for each site, particularly common between 14:00–23:00, when delphinids are shown to be vocally active. A known limitation of classification algorithms is that the output classes are restricted to those identified during training, the model cannot account for a new signal type, particularly one which presents such a similar structure to that of a signal of interest. By identifying this signal in the soundscape, we emphasise the importance of collecting broadband acoustic data to enable contextualisation of both the soundscape, and the detection algorithms output. Any new signal type identified within the soundscape will affect the confidence of detections output by the CNN, with lower probability scores per frame. By applying a threshold of 0.90 to the detection scores for outputting hourly presence data we reduced the overall false positives during April by 53%. The manual analysis revealed low delphinid site use of both Hyskier and Shiant Isles when the signal was present, suggesting a potential behavioural response. Further work will investigate the impact of this signal, its origin, and any long-term influence it may have on the dolphin populations of Scottish waters.

The dolphin populations on the west coast of Scotland are understudied with respect to other populations in the British Isles. Here we demonstrate that to uncover seasonal and diurnal trends in dolphin presence at each mooring within the COMPASS array the CNN provides a more robust insight to animal presence. We show that delphinid vocal activity is highest between 4 pm and 8am at each mooring, with higher densities of vocally active hours in the winter months (Figs 3 & 6), depicted by both detection algorithms. Seasonally we found variability in dolphin site use across the array; at Tolsta presence is high between November and January, dropping off during the summer seasons, but at Hyskier and Shiant Isles the dolphin 'season' appears earlier, with presence increasing from June through to November (Figs 3 & 6). The echolocation click is used primarily for hunting and navigation, suggesting these locations are important foraging sites throughout specific seasons, directing conservation and management input to locations of delphinid site use. As the CNN output higher performance metrics for all focal sites, and weather conditions, compared the C-POD, future work will make use of this detector to map distributions within the full COMPASS dataset, to attempt to understand dolphin movements in the west of Scotland.

## Conclusion

In the era of the Ocean Decade, long-term monitoring schemes for vulnerable species are increasingly important. We present performance statistics for two delphinid detection systems: a low-cost data logger with a standardized analysis process, and a CNN for rapidly disseminated raw acoustic data [49]. Through evaluation of these two methods, we present the CNN as open-source tool which is readily available for long-term monitoring of dolphin activity by exploiting the contents of broadband passive acoustic data. While the sensitivity of the C-POD is high, and false-positive rates remain low across diverse soundscape conditions, the logger fails to identify a significant proportion of detections, impacting its use as an accurate long-term tool for delphinid monitoring. When thresholds are used to control the CNN performance so that the *FPR* is equal to that of the C-POD then the CNN yields significantly better detection performance across all metrics. The benefit of a multi-sound source CNN as part of a PAM analysis toolkit permits automated signal extraction to occur in a timely manner and encourages researchers to gather large acoustic datasets for use in a wealth of ecological and environmental questions, important as we tackle the challenge of changing marine soundscapes.

## Supporting information

**S1 File.  Supplementary Material.pdf contains ancillary figures and tabulated data to support this manuscript.**
(PDF)

## Acknowledgments

We thank the creators of the COMPASS array for collecting such a valuable long-term PAM dataset used within this analysis.

## Author contributions

**Conceptualization:** Ellen L White, Paul R White, Jonathan M Bull, Denise Risch.

**Data curation:** Ellen L White, Denise Risch, Suzanne Beck, Susanna Quer.

**Formal analysis:** Ellen L White.

**Funding acquisition:** Paul R White, Jonathan M Bull, Denise Risch, Suzanne Beck, Susanna Quer.

**Investigation:** Ellen L White.

**Methodology:** Ellen L White, Paul R White, Jonathan M Bull.

**Supervision:** Paul R White, Jonathan M Bull.

**Validation:** Ellen L White.

**Visualization:** Ellen L White.

**Writing – original draft:** Ellen L White.

**Writing – review & editing:** Ellen L White, Paul R White, Jonathan M Bull, Denise Risch, Suzanne Beck, Susanna Quer.

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
