## [Decision Letter · Decision Letter 0]

7 Jan 2025

PONE-D-24-44181Evaluating the performance of automated detection systems for long-term monitoring of delphinids in diverse marine soundscapes.PLOS ONE

Dear Dr. White,

Thank you for submitting your manuscript to PLOS ONE. After careful consideration, we feel that it has merit but does not fully meet PLOS ONE’s publication criteria as it currently stands. Therefore, we invite you to submit a revised version of the manuscript that addresses the points raised during the review process.

We look forward to receiving your revised manuscript.

Kind regards,

Vitor Hugo Rodrigues Paiva, Ph.D.

Academic Editor

PLOS ONE

“This work was supported by the Natural Environmental Research Council [grant number NE/S007210/1]. The COMPASS project has been supported by the EU’s INTERREG VA Programme, managed by the Special EU Programmes Body. The views and opinions expressed in this document do not necessarily reflect those of the European Commission or the Special EU Programmes Body (SEUPB).”

Reviewers' comments:

Reviewer's Responses to Questions

**Comments to the Author**

1. Is the manuscript technically sound, and do the data support the conclusions?

Reviewer #1: Partly

Reviewer #2: Yes

2. Has the statistical analysis been performed appropriately and rigorously? 

Reviewer #1: Yes

Reviewer #2: Yes

3. Have the authors made all data underlying the findings in their manuscript fully available?

Reviewer #1: Yes

Reviewer #2: Yes

4. Is the manuscript presented in an intelligible fashion and written in standard English?

Reviewer #1: Yes

Reviewer #2: Yes

5. Review Comments to the Author

Reviewer #1: 1. Could the authors clarify the novelty of this study to highlight its unique contributions? Since the CNN model used here was originally designed in another work (citation [44]), is the novelty primarily in its application to a new dataset?

2. In section 3.1-3.4, the authors compare the CNN with Manual analysis and the C-POD, but several issues need attention:

In C-POD, the KERNO classifier detects click train based on parameters such as intensity, duration, frequency content and inter-click intervals (ICI). To ensure fair comparisons, the Manual Analysis and CNN should also output click train detections specifically.

2.1 CNN: The authors state (Lines 168-169) that the CNN classifies chunks as containing delphinid clicks if the chunks include both click trains and burst pulses, but burst pulses are distinct from click trains. Additionally, the authors state on Line 175 that the CNN model prioritizes whistles over clicks. To align with the aim of detecting click trains, either the CNN model from [44] should be fine-tuned, or a new model should be trained specifically for the exclusive detection of click trains. Without this adjustment, the comparisons presented in Sections 3.1–3.4 lack validity. Furthermore, this adjustment could serve as a significant contribution or highlight the novelty of this study.

2.2 Manual analysis: (Page 8, line 151) The method for identifying click trains in the Manual Analysis needs further clarification. It would be helpful to align the manual analysis with the approach used by the C-POD --considering parameters such as intensity, duration, frequency content, and inter-click intervals (ICI), rather than relying solely on the number of clicks in a click-series.

Since the aim of this work is to develop an automated tool for extracting species presence, all types of clicks (e.g. click trains, burst pulses, buzzes, and regular echolocation clicks) can be used. Therefore, the C-POD should be tuned to detect all these signals, not just click trains, and the CNN model from [44] should be adjusted to prioritize click detection over whistles.

In summary, any comparisons should be made based on consistent outputs across the different methods to ensure validity and reliability.

3. The identification of vessels (both manual and CNN) is not utilized in this study. To avoid confusion for readers, please consider removing related descriptions.

4. This work lacks a background survey on the application of Deep Learning to marine mammal sound detection or classification. While the authors provide reasons for using CNN on Page 5, Line 83, could they also include examples of other state-of-the-art Deep Learning applications to further support their choice of CNN?

5. Abstract: please consider merging the two paragraphs into a single cohesive paragraph to provide a concise summary of the work.

6. page 3, line 44: missed a period after ` offline systems’.

7. Page 4 lines 56-57: Does this sentence means: the C-POD detects individual sound clicks while operating (on-line), and later, during offline processing, those clicks are grouped together into click trains based on their characteristics.

The authors mention that the C-POD performs online detection; however, the sudden introduction of offline processing could confuse readers. It would be helpful to provide a clearer explanation of the processing steps involved in the C-POD detection workflow, including how and when the transition from online detection to offline processing occurs.

8. Page 4, line 60: There are two ‘for which’ at the beginning of this line.

9. Considering adding a sentence at the end of the Introduction to outline the structure of the manuscript. This will help guide readers through the subsequent sections.

10. In Figure 2 or in the text, please provide an example of the labels used for CNN, similar to the numeric labels presented for Manual Validation and C-POD. This would ensure consistency and help readers better understand the comparison.

11. Page 8 line 157: Please provide a detailed description of the CNN architecture used in this work. Relying on references to another paper for such essential information could hinder readers' understanding of your methodology.

12. The authors have provided the threshold for both whistle and click detections, but have not specified the threshold used to determine the presence of clicks. Please include this information for clarity.

13. Page 8 line 174: Please clarify the meaning of ‘frame’ in the context of this work. Is it synonymous with ‘chunk,’ or does it refer to a different concept?

14. Page 9, lines 182-186: Are there existing statistics in the literature regarding the performance of the C-POD in detecting click trains? If so, please include this information to strengthen the rationale for selecting this method for comparison.

15. For Table 2, please consider adding an additional row to show the overall number of hours for each period. This would provide a clearer summary of the data.

16: Page 10 line 215: The sum of 532, 839 and 566 equals 1937, not 1936. Please double-check this calculation and verify all numerical values in tables for accuracy.

17: Page 10 line 203: The description of FPR in this line is incorrect. FPR refers to the proportion of actual negatives that are incorrectly classified as positives. Please revise this definition for accuracy.

18. Page 11 lines 218-219: Could the authors clarify how the proportions of dph were calculated? Please also double-check that all proportions provided are accurate.

19. Page 15 line 327, page 16 line 333: In the parenthesis on both lines (Table, Figure 5), please specify which sub-figure in Figure 5 is being referred to for the discussion to ensure clarity.

20. Page 16, line 345: the authors mention that the CNN generated a single label for each audio data chunk. However, on page 8 Line 168, it is stated that the network classifies each chunk into one of four labels: ambient noise, delphinid whistles, delphinid clicks (click trains and burst pulses), or vessel noise. This inconsistency is confusing and needs clarification.

Reviewer #2: The paper "Evaluation the performances of automated detection systems for long-term monitoring of delphinids in diverse marine soundscapes" is well written, interesting and with a solid conclusion. It is a useful paper that will help scientists and conservationists in their choice of a monitoring device.

However, I have one serious objection (why say you are only comparing detection software and not the recording device ?), and think the presentation of the manuscript could be improved to better the general clarity of the work, and help further the future users of these instruments. I will first mention some general suggestions and then go into detailed comments.

***** General comments :

* On the general purpose (and title) : to my judgement, the comparison not only involves two detection techniques, but also two recording devices - it is difficult to separate them, given the integrated nature of the C-POD . CNN analyze on data gathered with a poor recording device would probably have a different efficiency. This 'objection' does not lessen the utility of the work, it's rather a suggestion that the study should clearly mention hardware as well as software in the comparison of the performance.

* Use of metrics : the metrics used are generally well defined (with some exception, see detailed comments), but may be to numerous, at the risk of losing the reader in a forest of parameters without always explaining the interest of each. I would globally advice to use fewer metrics and stick to them in all comparisons, justifying which is interesting in each case. This is already partly done in the paper but could maybe be more systematic.

* The comparison is well detailed in the 'results' part (maybe with too much details?) but very few explanations are given to explain the trends. The high level of false positives in the whistle detection is explained by a deeper insight into the manually revised data. The same should be done for the other main results : why is the C-POD so little sensible ? Can it simply be because of the recording device sensibility (and not the detection software) ? From my experience, individual C-POD can have different sensibilities, especially if 'old'. You could perhaps check the signal to noise ratio of the manual detections in some cases of lost information, for instance. At high frequency, the level of noise is sometimes low and sensibility of the recording device can be quite important. I would maybe suggest that 100 m is rather deep for click detection, given the directionnality of the clicks and of the sensors at high frequency ?

* The figures quality should be improved (maybe some of them could be skipped without changing the quality of the paper).

***** Detailed comments :

* Abstract :

l.4 "the work compares two methods" : as already mentioned in the general comment, I think what is compared is "recording device + method", the two cannot be separated. Thus I would mention the Soundtrap device as part of the general result.

* Introduction :

l.25 ref (1) does not seem to be exactly to the point ?

l.26 "to effectively protect marine these species": not very clear

l.27 ref(3) is about invertebrate and noise , not really habitat use and distribution ?

l.34 ref (15) and (16) are only about signature whistles, it could be more general to illustrate your point ?

l.35 "signals between 20-100 kHz" : what about narrow band high frequency (NBHF) delphinids ? Their clicks go up to 135 kHz or more ... Even if you are not concerned in Scotland, the definition should not exclude them.

l.44 "systems Off-line" -> "systems. Off-line "

l.52 "C-POD" presentation : you should definitely explain here that C-POD are outdated, and mention F-POD. Since a lot of scientists or ONG still use C-POD, it takes nothing away from the interest of your study.

l.94 the CNN has been trained (partly) on data from the same project in one of the places (Tolsta), hasn't it ? You should maybe mention this here.

* Methods :

l.113 In my copy, fig 1 has a very low quality.

l.143 In my copy, fig 2 has a very low quality.

l.160 cf my comment on line 94

l.170 : I couldn't see where you mentioned what probability threshold you use, and why (because it was the one you used in your previous work?). At the end you mentioned a 0.9 threshold, but I seem to understand it was not your first choice ?

l.175 "where whistles were prioritised over clicks due to a shorter temporal length" : I don't really understand this sentence.

l.177 "a dph determined for ..." -> "a dph is determined for ..."

l.179 which threshold ?

l.193 and supplementary material : OK

l.198 : "These calculation use the manual labels as ground truth" I would mention here the remark of l. 384 and following, that the "ground truth" is counted only on 1/3rd of the total time ! It is rather counter-intuitive, though rather well justified in l.386.

l. 199 : I would mention (since these are notions that you will use later) that Recall is also called Sensitivity and it is also the TPR (True Positive Rate)

l.203 : "the FPR is the proportion of the positive detections that are correct" is uncorrect

l.204 : FNR should be (better) explained. It is also 1-Recall, which means, I think, that it is not very useful to mention it along with Recall.

l.208 : "manually" -> "manual"

* Results

l.211 : Mention (in the 3.1 title?) that here you are comparing click detection only (and not delphinid detection). It is written in l.213, but not very obvious when you read it for the first time

l.221 : table 2 : you could maybe mention the % of TP, to make the reading easier (E.g. N_{TP} = 452 (46%) )

l.226 : it is interesting to mention the processing time of each method. CNN has been trained in another study but with the same kind of data and environment, maybe you should mention it ? It would be interesting, also, to mention the volume of data for each case ? Also, maybe give directly the total time, not "by week of data" so the reader doesn't have to look back how many weeks you had and do the multiplying for himself ;)

l.231 : C-POD in this case has a high precision but low recall (more classically paired with precision than FNR)

l.238 : FNR is 1-R so it is not really indispensable to put it in this table, which is already rather large.

l.247-250 : a relevant comment

l.263 : In my copy, fig 3 has a very low quality.

l.265 : why smooth the data ?

l.275 : "correlation was weakest at Shiant Isles" : could you explain why (or in the discussion part) ? It seems very low indeed.

l.291 : TPR has not been defined (=R)

l.293 : FNR is a bit redundant here, isn't it (=1-TPR = 1-R) ?

l.296 : In my copy, fig 4 has a very low quality. You could use a more standardized scale (0-1, except maybe for FPR which is usually low in unbalanced populations).

l.311 : Could you maybe justify why you chose these metrics, not the same as before? Precision was not considered useful in this case ?

l.320 : In my copy, fig 5 has a very low quality. I would definitely use the same scales for the different sites.

l.329 : Storm may not be of great influence on the performance of the devices then ? Did you check in the data to see how 'storm' was translated into noise? White noise, or colored, constant, or with interruptions? It's not obvious how a storm is recorded at more than 50 m deep.

l.331-338 : It seems to me the comparisons here are a bit confusing, you are using different metrics for the different methods.

l.340 : this conclusion seems rather in contradiction with the sentence of l.328-329 ?

l.349 : here, the threshold is mentioned.

l.343 : General comment on part 3.5 : This part is rather short compared with the previous one and could maybe be a little more detailed (or removed altogether if you want to stick to clicks detections). The very high numbers of false positives, not only in April (table 6) are rather shocking (they tend to claim that dolphins were present 88 % of the time in overall, with 100 % in nov in 2 sites). During the storms, the number is especially high, triplicating the detections). This result should be addressed in the first place, I think, with the other comments coming on later. Why none of the metrics that were previously described are used in this part ?

l.352 : In my copy, fig 6 has a very low quality.

* Discussion

l.377-382 : This paragraph is a bit repetitive and not really necessary in my opinion.

l.386 : see my comment on l. 198

l.391-392 : I don't really agree with this comment, I think a lot can be explained of the performances of each when going more in detail into the data (like it is done later, L.440 for the FPR of whistle detection).

l.400 : "unseen acoustic data" see my comment on l.94

l.404 : The whistle detection improved the correlation coefficient but the very high rate of FP is alarming.

l.416 : "when applying ..." I don't get it : was not 0.9 the threshold you chose in the first place(l.349) ?

l.427 : there are other studies which did not find the same result, I think? Could it be due to this individual C-POD performing poorly? To the set-up, not favorable for this instrument? To the fact that the C-POD is performing better for NBHF species (Narrow band High frequency) ? It could be interesting to include these elements to your discussion, for a more balanced (and thus more useful) conclusion.

l.428 : see my comment l.35 ...

l.428 : yes. I think the sensitivity of the instruments, as well as the detection ranges, could have been introduced higher in the paper, I feel they are constitutive of the comparison.

l.433 : ref 31, not 301

l.440 : not only in April ...

l.440 : this paragraph is interesting, you could maybe show the signal in the suppl. material?

l.452 : I am still lost on the threshold that was applied in the first place.

l.461-462 : maybe "vocal" activity is not the correct term for clicks?

* Bibliography

l.542 : could you give Editor ?

l.604 : idem

l.688 : idem

6. PLOS authors have the option to publish the peer review history of their article (what does this mean? ). If published, this will include your full peer review and any attached files.

**Do you want your identity to be public for this peer review?** For information about this choice, including consent withdrawal, please see our Privacy Policy .

Reviewer #1: No

Reviewer #2: No

---

## [Author Response · Author response to Decision Letter 1]

26 Feb 2025

We are grateful for the detailed and thoughtful reviews provided on our manuscript. The reviewers’ comments were comprehensive however we wish to stress that the changes to the manuscript are minor, and our major findings are unchanged. We believe that the manuscript has been improved by the review process and appreciate the thought-provoking comments provided by each reviewer. We have attached a document (word document) containing our full response to each of the reviewers comments.

---

## [Decision Letter · Decision Letter 1]

25 Mar 2025

PONE-D-24-44181R1Evaluating the performance of automated detection systems for long-term monitoring of delphinids in diverse marine soundscapes.PLOS ONE

Dear Dr. White,

Thank you for submitting your manuscript to PLOS ONE. After careful consideration, we feel that it has merit but does not fully meet PLOS ONE’s publication criteria as it currently stands. Therefore, we invite you to submit a revised version of the manuscript that addresses the points raised during the review process.

We look forward to receiving your revised manuscript.

Kind regards,

Vitor Hugo Rodrigues Paiva, Ph.D.

Academic Editor

PLOS ONE

Reviewers' comments:

Reviewer's Responses to Questions

**Comments to the Author**

1. If the authors have adequately addressed your comments raised in a previous round of review and you feel that this manuscript is now acceptable for publication, you may indicate that here to bypass the “Comments to the Author” section, enter your conflict of interest statement in the “Confidential to Editor” section, and submit your "Accept" recommendation.

Reviewer #1: (No Response)

Reviewer #2: All comments have been addressed

2. Is the manuscript technically sound, and do the data support the conclusions?

Reviewer #1: Yes

Reviewer #2: Yes

3. Has the statistical analysis been performed appropriately and rigorously? 

Reviewer #1: Yes

Reviewer #2: Yes

4. Have the authors made all data underlying the findings in their manuscript fully available?

Reviewer #1: Yes

Reviewer #2: Yes

5. Is the manuscript presented in an intelligible fashion and written in standard English?

Reviewer #1: Yes

Reviewer #2: Yes

6. Review Comments to the Author

Reviewer #1: 1. The quality of most figures in my copy is really low. Are they vector figures? If not, please convert them to vector figures for better quality.

2. Some of the line numbers in your responses do not match the actual line numbers in your revised PDF.

For example: “Line 103, on page 5 now reads: One widely adopted ML approach makes use of Convolutional Neural Networks (CNNs) …” this sentence is in line 93. For example: “I have now removed the second ‘for which’ from line 80”, but I found the ‘for which’ in line 72.

For example: “Line 231: During post-processing a threshold can be…” is in line 223 of your revised pdf.

3. There are still two paragraphs in the abstract.

4. Line 101, ‘This section outlines the methodology use, beginning….’, please change ‘use’ to ‘used’.

5. Adding a sentence about the overall structure of the manuscript at the end of the introduction is to give readers a clear picture of your manuscript. You can introduce each section at the beginning of that section, but please provide only a summary rather than listing details of all subsections.

6. For figure 2, please avoid adding many texts in the caption -- a picture is worth a thousand words. If possible, please add an example of the labels used for

CNN in the figure 2, similar to the numeric labels presented for Manual Validation and C-POD.

7. Let’s revisit to the novelty of this manuscript. The purpose of your study is to detect the presence or absence of delphinids, and the objective of this study is to demonstrate the broader capabilities of deep learning and highlight how an off-the-shelf network—without extensive fine-tuning—can still produce meaningful insights into species presence.

I believe this work underscored the importance of the study presented in “Automated detection of marine sound sources with a convolutional neural network”, but it still lacks sufficient novelty for a new manuscript. The comparison between the performance of your CNN model and the C-POD is valuable and represents a large amount of data processing and effort. I truly appreciate your contribution to this field, but, comparing an existing CNN model (already published) with a device (C-POD) without a clear understanding of the specific detection methods used in the device is not sufficient for publication.

Reviewer #2: The authors have carefully considered both reviewers suggestions.

They decided to follow most of the recommendations, if not all, and explained their choices. I don't agree with all of these decisions but consider their work should be published as such. Figures still have a rather poor resolution in my version.

7. PLOS authors have the option to publish the peer review history of their article (what does this mean? ). If published, this will include your full peer review and any attached files.

**Do you want your identity to be public for this peer review?** For information about this choice, including consent withdrawal, please see our Privacy Policy .

Reviewer #1: No

Reviewer #2: No

---

## [Author Response · Author response to Decision Letter 2]

4 Apr 2025

Dear Editor,

We are grateful for the further comments supplied by the reviewers and thank Reviewer 2 for their support in the publication of our manuscript. Reviewer one supplied further comments, which we have responded to in detail below (1-6).

Yours Sincerely,

Dr Ellen White

School of Ocean and Earth Sciences

University of Southampton

1. The quality of most figures in my copy is really low. Are they vector figures? If not, please convert them to vector figures for better quality.

Thank you for raising the concern regarding figures again, however the submitted figure quality is high and is being reduced during the pdf conversion as per the review portal. The final published article will not contain the low quality as seen here.

2. Some of the line numbers in your responses do not match the actual line numbers in your revised PDF.

For example: “Line 103, on page 5 now reads: One widely adopted ML approach makes use of Convolutional Neural Networks (CNNs) …” this sentence is in line 93. For example: “I have now removed the second ‘for which’ from line 80”, but I found the ‘for which’ in line 72.

For example: “Line 231: During post-processing a threshold can be…” is in line 223 of your revised pdf.

I apologise for the miscommunication regarding line numbers within the feedback. I really appreciate the efforts that went into your review and have addressed each point accordingly. The line numbers will have shifted with multiple authors editing the text in response to each point and a few inconsistencies will have remained. In future I will strive to ensure this does not happen again.

3. There are still two paragraphs in the abstract.

The paragraphs in the abstract have been condensed into one body of text (Lines 2-18).

4. Line 101, ‘This section outlines the methodology use, beginning….’, please change ‘use’ to ‘used’.

The use of ‘use’ is changed to ‘used’ in line 102.

5. Adding a sentence about the overall structure of the manuscript at the end of the introduction is to give readers a clear picture of your manuscript. You can introduce each section at the beginning of that section, but please provide only a summary rather than listing details of all subsections.

I thank the reviewer for their continued efforts at improving the manuscript. The final paragraph of the introduction does currently provide an overview of the analyses conducted within the manuscript, in order. If the reviewer has further suggestions to improve this I would welcome them, but do not see how this can be further enhanced beyond the existing text:

‘In this work we conduct the first empirical evaluation of the performances of a CNN, applied to broadband acoustic data recorded on a SoundTrap 300HF (Ocean Instruments), and the C-POD on the task of delphinid detection off the west coast of Scotland (Figure 1). An open-source multi-sound source CNN [49] is used without any network adaptation to the test data. We compare the output of both algorithms, across diverse acoustic conditions, against manually labelled recordings to evaluate their suitability in the standard processing toolkit for long-term delphinid monitoring’.

The introductory section header for the methods section (Line 102 – 105) has been condensed to:

‘This section outlines the methodology used, beginning with a detailed summary of the data acquisition process (Section 2.1), describing the analysis of acoustic data through each approach: a) manual, b) the CNN, and c) the C-POD (Section 2.2) and the evaluation metrics used in subsequent sections to assess the performance of each approach (Section 2.3).’

Given the vast detail within the results section, I feel the current introductory paragraph is needed to guide the reader through the structure of this sentence. This detail is difficult to include higher up in the introduction as the reader will require the context of the methods section to follow it. I have tweaked it (Lines 262 – 269) to read:

‘This section evaluates the performance of the C-POD and the CNN, applied to broadband acoustic data, in detecting delphinid presence across diverse underwater soundscapes. This analysis is based on 3,000 hours of acoustic data collected throughout 2019 in western Scottish waters. First, we conduct an overall comparison of click detections (Section 3.1). The output of each detector is then examined to identify temporal and seasonal patterns (Section 3.2). Sections 3.3 and 3.4 compare detector performance across varying ambient noise conditions using ROC curves and precision-recall metrics. Finally, Section 3.5 incorporates whistle detections output by the CNN to evaluate the benefits of monitoring delphinids using multiple signal types.’

6. For figure 2, please avoid adding many texts in the caption -- a picture is worth a thousand words. If possible, please add an example of the labels used for

CNN in the figure 2, similar to the numeric labels presented for Manual Validation and C-POD.

Regarding the caption for Figure 2, the detail provided was in line with the request of Reviewer 2 to enhance overall clarity. We feel the caption succinctly summarises the complex figure, providing clarity to the reader outside of the main text. I have edited it slightly to reduce the word count, but feel that the current text is justified against the level of detail in the figure (Lines 165 – 172):

‘Figure 2. Schematic depiction of the data collection and subsequent analysis workflow used for the determination of delphinid detection positive hours (dph) by a) Manual, b) CNN, and c) CPOD detection. Manual analysis applies a positive or negative label for the presence of delphinid clicks and delphinid whistles. Audio data is fed through the CNN which outputs one label per input frame (3 seconds); Ambient noise, delphinid whistle, delphinid click or vessel noise, and the total frames per class are summed per file to provide hourly presence/absence of clicks. The C-POD outputs detection positive minutes for delphinid clicks, for each minute of the recording hour, with the total sum of positive minutes converted to positive or negative detection hour.’

I thank the reviewer for their suggestion within Figure 2. In the CNN pipeline, I show the labels ‘Click/Whistle/Vessel’ as the signals which the network can detect. This is line with what is shown for Manual Validation and C-POD and so I have not added any additions to this figure. If the reviewer means to add [1,1] similar to manual validation we do not think this is appropriate as there are three label options for the CNN and to demonstrate these as binary arrays would require adding: [1,0,0], [0,1,0], [0,0,1]This is detail that we do not feel is necessary within this figure.

7. Let’s revisit to the novelty of this manuscript. The purpose of your study is to detect the presence or absence of delphinids, and the objective of this study is to demonstrate the broader capabilities of deep learning and highlight how an off-the-shelf network—without extensive fine-tuning—can still produce meaningful insights into species presence.

I believe this work underscored the importance of the study presented in “Automated detection of marine sound sources with a convolutional neural network”, but it still lacks sufficient novelty for a new manuscript. The comparison between the performance of your CNN model and the C-POD is valuable and represents a large amount of data processing and effort. I truly appreciate your contribution to this field, but, comparing an existing CNN model (already published) with a device (C-POD) without a clear understanding of the specific detection methods used in the device is not sufficient for publication.

While we appreciate Reviewer One's concerns regarding novelty (7), we respectfully suggest that our application of an existing neural network from published literature represents a meaningful contribution. Specifically, our work demonstrates—for the first time in this context—how deep-learning algorithms can be effectively deployed to extract species presence from acoustic recordings made in unseen marine soundscapes. This approach significantly advances accessibility for non-technical users in the field.

Our work demonstrates a rigorous comparison between established commercial data logging systems (C-POD) and emerging machine learning approaches. This comparison provides timely and valuable insights for the research community as machine learning tools are becoming standard within this domain. We provide a detailed characterisation of detection biases for both approaches, offering practitioners a nuanced understanding of each methods strengths and limitations. We expect that our analyses will enable informed decision-making about monitoring strategies based on specific research questions and environmental contexts.

The reviewers concern regarding the C-PODS proprietary algorithms we find misplaced. We note that if understanding the internal algorithms are a prerequisite for robust comparison it would invalidate a substantial body of published literature which relies on C-POD data. Our approach evaluates each system on their observable performance characteristics, which is standard practice when comparing emerging technologies. We hope that our research is accessible to researchers from a range of scientific disciplines, and we provide baseline metrics on each method of detection to educate the community on considerations they should address when designed passive acoustic monitoring projects for monitoring toothed whales.

---

## [Editor Report · Decision Letter 2]

15 Apr 2025

Evaluating the performance of automated detection systems for long-term monitoring of delphinids in diverse marine soundscapes.

PONE-D-24-44181R2

Dear Dr. White,

We’re pleased to inform you that your manuscript has been judged scientifically suitable for publication and will be formally accepted for publication once it meets all outstanding technical requirements.

Kind regards,

Vitor Hugo Rodrigues Paiva, Ph.D.

Academic Editor

PLOS ONE
---

## [Editor Report · Acceptance letter]

PONE-D-24-44181R2

PLOS ONE

Dear Dr. White,

I'm pleased to inform you that your manuscript has been deemed suitable for publication in PLOS ONE. Congratulations! Your manuscript is now being handed over to our production team.

Kind regards,

on behalf of

Dr. Vitor Hugo Rodrigues Paiva

Academic Editor

PLOS ONE